



# The Spatio-Temporal Visualization Tool HMMLVis in Renewable Energy Applications

Rainer Wöß[1], Kateřina Hlaváčková-Schindler[1], Irene Schicker[2], Petrina Papazek[2], and Claudia Plant[1]

[1]Research Group Data Mining and Machine Learning, Faculty of Computer Science, University of Vienna, Währingerstrasse 29, 1090 Vienna, Austria
[2]GeoSphere Austria, Hohe Warte 38, 1190 Vienna, Austria

**Correspondence:** (RW) rainer.woess@univie.ac.at, (KHS) katerina.schindlerova@univie.ac.at, (IS) irene.schicker@geosphere.at, (PP) petrina.papazek@geosphere.at, (CP) claudia.plant@univie.ac.at

**Abstract.** In this work, we present HMMLVis, an original visualization tool for multivariate Granger causal inference. More precisely, for heterogeneous Granger causality to infer causal relationships in time-series following an exponential distribution. HMMLVis is easy to use and can be applied in any scientific discipline exploring time series and their relationships. In this paper, we focus on climatological and meteorological applications. The visualization tool is demonstrated on different types
of applications related to meteorological events on the upper/lower tails of the respective distributions using a renewable energy (wind, PV), air pollution, and the EUMETNET postprocessing benchmark data set (EUPPBench) and different temporal horizons. We demonstrate that the HMMLVis method and visualization depicts the known causal and detects causal relations in the temporal dependencies which are additional important information for the respective cases. We believe that HMMVis as an interpretable visualization tool will serve climatologists or meteorologists and in this way it will contribute to knowledge
discovery in these scientific fields.

## 1 Introduction

In large-scale complex dynamical systems such as meteorology / climatology and fields and system affected by meteorological and climatological conditions such as renewable energy production systems, replicated interventional experiments are rarely feasible or ethically problematic. The rapidly increasing amount of observations and numerical-dynamical generated data, e.g.
reanalysis or numerical weather prediction data, opened the very fast developing field of data-driven deep learning forecasting methods. In data-driven predictions, the interplay of features and causality gain more importance in improving the forecast quality and skill. Applying observational causal discovery methods can help to improve predictions by their ability to quantify potential causal dependencies from time series data without the need to intervene in systems (e.g. manipulate part of a system and infer relationships from the consequences).

In contrast to deep learning, which mainly focus on prediction and classification, causal inference methods aim at discovering and quantifying the causal dependencies of the underlying system. Although interpreting deep learning models is an active area of explainable AI, extracting the causes of particular phenomena, e.g., extreme flooding, from a deep learning structure with multiple layers is usually not possible. Conversely, causal inference methods can contribute in theoretical understanding of the



underlying system and, combined with deep learning methods, can help improving predictions and classifications. Recently
more attention was brought to causal methods in climatology, as Runge et al. (2023) have introduced a taxonomy of research
questions. These are types of expert assumptions and properties of the available time series data to provide a causal language in
which researchers can define their study questions. To make these causal methods even more accessible, they could be provided
in visual tools for researchers to perform experiments on observation data.

Our work presents a visual tool in HMMLVis, which uses a method extending Granger causality, named HMML, to in-
fer causal relationships in time-series data. Granger causality addresses causal relationships quantitatively by comparing the
prediction error of time-series models given the inclusion or exclusion of another variable's model.

In its original linear form as introduced in Granger (1969), criticism arose as Granger causality does not take counterfactuals
into account (Mannino and Bressler, 2015; Maziarz, 2015) and does not fulfill the causal sufficiency, i.e. non-existence of a
hidden common cause (Spirtes, 2010). Further, the regression equations reflect only correlations and the Granger test detects
only "predictive causality". In defense of the method, Granger (1988) wrote: "Possible causation is not considered for any
arbitrarily selected group of variables, but only for variables for which the researcher has some prior belief that causation is, in
some sense, likely." In other words, drawing conclusions of the existing causal relation between a time series and its direction
is only possible if some theoretical knowledge of mechanisms connecting the time series is accessible. Granger stressed that a
proper use of Granger causality would require to condition on all relevant variables in the world. In fact, conditioning on the
"whole universe" is a deficiency not only of the Granger causal model but of most models used for multivariate causal inference,
including the structural causal models (SCM, e.g. Spirtes et al. (2000)) since not always all variables (direct, confounding,
latent) are known and using such a high degree of parameters would blow up every modeling system. This condition is, in
practice, relaxed by an assumption that the set of all relevant variables is known. Thus, the selection of relevant variables is a
necessity and extremely crucial step and cannot be done without domain experts. Therefore, here we assume a set of relevant
parameters with the assumption of causal relationship between them provided by expert domain knowledge. Granger causal
inference can also answer certain counterfactual questions when applied to observational data which can be pooled or data of
seasonal nature, e.g. of climatological origin.

Thus it is generally accepted that Granger causality does not capture all aspects of causality but enough to be worth consid-
ering for empirical test (Singh and Borrok, 2019). In observational data, namely meteorological and climatological time series,
where replicated interventional experiments are hardly feasible, Granger causality and its non-linear and graphical variations
can provide a valuable insight into the the temporal relationships about variables. Thus, besides the rapid development of other
causal inference methods, Granger causality and its non-linear and multivariate versions still play an important role in Earth
system sciences, as demonstrated by the unremitting publications this field, e.g. on prediction in photovoltaic data set (Shan
et al., 2023), environmental quality assessment (Celik and Alola, 2023), or in general renewable energy production (Dumrul
et al., 2023).

In this work, we present our original visualization tool for Granger causal inference, HMMLVis, namely for the case of
heterogeneous Granger causality (Behzadi et al., 2019). The tool is demonstrated on different types of applications related
to meteorological events, namely renewable energy production by photovoltaic and EUMETNET postprocessing benchmark



data set (EUPPBench) from Demaeyer et al. (2023). Further, other utilization of HMMLVis is discussed for air polution data,
semi-synthetic wind production data for a selected wind farm location and for semi-synthetic photovoltaic power generation.

The paper is organized as follows. Section 2 presents heterogeneous Granger causality and Method HMML and Section
3 describes related work. Our HMMLVis method is introduced in Section 4. Workflow and description of the visualisation
tool of HMMLVis is explained in Section 5. Section 6 discusses utilisation of HMMLVis in several climatological or energy
-production applications. Our conclusions are summarized in Section 7.

## 2   Heterogeneous Granger Causality and Method HMML

The original bivariate concept of causality defined by Granger (1969) can be extended to the multivariate case, i.e. for $p > 2$
time series and a time lag $d \geq 1$, indicating the maximum number of lagged observations included in the model, the so-called
model order. This model order can be selected via an information theoretic criteria such as the Bayesian or Akaike information
criterion. For $p$ time-series $\boldsymbol{x}_1, .., \boldsymbol{x}_p$ the vector auto-regressive (VAR) model is:

$$x_i^t = \boldsymbol{X}_{t,d}^{Lag} \boldsymbol{\beta}_i' + \epsilon_i^t \tag{1}$$

where $\boldsymbol{X}_{t,d}^{Lag} = (x_1^{t-d}, .., x_1^{t-1}, .., x_p^{t-d}, .., x_p^{t-1})$, $\boldsymbol{\beta}_i'$ is the transposition of the matrix $\boldsymbol{\beta}_i$ of the regression coefficients and $\epsilon^t$
the error Behzadi et al. (2019). It is stated that the time-series $\boldsymbol{x}_j$ Granger-causes the time-series $\boldsymbol{x}_i$ for lag $d$ if and only if at
least one of the $d$ coefficients in row $j$ of $\boldsymbol{\beta}_i$ is non-zero. Thus, to detect causal relations, coefficients of the VAR model need
to be estimated. This problem can be solved by a penalization approach of the order $d$, e.g. by using Lasso Tibshirani (1996),
often also referred to as variable selection method.

Multivariate Granger causality among time series from Eq. (1), as an instance of graphical causal models, Glymour et al.
(2019), assumes that random error time series follow Gaussian distributions with zero mean and constant deviation. In many
applications however, these assumptions do not hold and using a graphical Granger model can lead to an inaccurate causal
inference results. Profiting from the framework of the generalized linear models (GLM) introduced in Nelder and Wedderburn
(1972), Behzadi et al. (2019) proposed a general model to detect Granger-causal relations among $p \geq 3$ number of time series
which follow a distribution from the exponential family (where Gaussian distribution is a special case). The relation among the
response variable and the covariates in a regression is not linear anymore but defined by a so-called link function $\boldsymbol{\eta}$, a monotone
twice differentiable function depending on the concrete distribution functions from the exponential family.

The heterogeneous graphical Granger model (HGGM), Behzadi et al. (2019), considers time series $\boldsymbol{x}_i$ which follow a
distribution from the exponential family using a canonical parameter $\boldsymbol{\theta}_i$. The generic density form for each $\boldsymbol{x}_i$ can be written
as:

$$p(\boldsymbol{x}_i | \boldsymbol{X}_{t,d}^{Lag}, \boldsymbol{\theta}_i) = h(\boldsymbol{x}_i) \exp(\boldsymbol{x}_i \boldsymbol{\theta}_i - \eta_i(\boldsymbol{\theta}_i)) \tag{2}$$

where $\boldsymbol{\theta}_i = \boldsymbol{X}_{t,d}^{Lag}(\boldsymbol{\beta}_i^*)'$, with $\boldsymbol{\beta}_i^*$ being the optimum, and $\eta_i$ is a link function corresponding to time series $\boldsymbol{x}_i$. The HGGM
uses the idea of generalized linear models and applies them to time series in the following form





$$x_i^t \approx \mu_i^t = \eta_i^t(\boldsymbol{X}_{t,d}^{Lag}\boldsymbol{\beta}_i') = \eta_i^t(\sum_{j=1}^{p}\sum_{l=1}^{d}x_j^{t-l}\beta_j^l) \tag{3}$$

for $x_i^t$, $i=1,\ldots,p,t=d+1,\ldots,n$ each having a probability density from the exponential family; $\boldsymbol{\mu}_i$ denotes the mean of $\boldsymbol{x}_i$ and $var(\boldsymbol{x}_i|\boldsymbol{\mu}_i,\phi_i)=\phi_i v_i(\boldsymbol{\mu}_i)$ where $\phi_i$ is a dispersion parameter and $v_i$ is a variance function dependent only on $\boldsymbol{\mu}_i$; $\eta_i^t$ is the t-th coordinate of $\boldsymbol{\eta}_i$. The causal inference in (3) can be solved as a maximum likelihood estimate for $\boldsymbol{\beta}_i$ for a given lag $d>0$, $\lambda>0$, and all $t=d+1,\ldots,n$ with added adaptive lasso penalty function Behzadi et al. (2019). One can state that time series

$\boldsymbol{x}_j$ Granger–causes time series $\boldsymbol{x}_i$ for a given lag $d$, and denote $\boldsymbol{x}_j \rightarrow \boldsymbol{x}_i$, for $i,j=1,\ldots,p$ if and only if at least one of the $d$ coefficients in $j-th$ row of $\hat{\boldsymbol{\beta}}_i$ of the penalized solution is non-zero, see Behzadi et al. (2019).

The concept explained in this paragraph replaces the solution via $p$ penalized linear regression problems by formulating the feature selection as a combinatorial optimization problem, as it was done in Hlaváčková-Schindler and Plant (2020a) for the multivariate Granger causal model with Gaussian time series and in Hlaváčková-Schindler and Plant (2020b) for the multi-

variate Granger causal model with time series from the exponential family. The second method uses the information theoretic criterion "minimum message length" (MML), introduced by Wallace (2005) for general inference problems, to determine causal connections in the model, improving the results especially for short time-series. This will be the focus of the following work, as e.g. in the wind related data set, we initially analyze measurements of 96 samples.

The MML principle is a formal information theory restatement of Occam's razor: Even when models have a comparable

goodness-of-fit to the observed data, the one generating the shortest overall message is more likely to be correct (where the message consists of a statement of the model, followed by a statement of data encoded concisely using that model). The statistical version of the MML principle constructs a description in terms of probability functions and some prior knowledge of the parameter vector. MML seeks the model that minimizes this trade-off between model complexity and model capability. In the type of MML considered in Hlaváčková-Schindler and Plant (2020b) and in this study and application, the parameter space

$\boldsymbol{\theta}$ for the statistical model $p(.|\boldsymbol{\theta})$ is decomposed into a countable number of subsets and associated code words for members of these subsets. The parameter $\boldsymbol{\theta}$ in the MML criterion corresponds to the maximum likelihood estimates of the regression coefficients and the dispersion coefficient of the target time series. The regression problem is expressed for $i=1,\ldots,p$ via incorporation of a subset of indices of regressor variables, denoted by $\boldsymbol{\gamma}_i \subset \{1,..,p\}$ and $k_i=|\boldsymbol{\gamma}_i|$ into (3)

$$x_i^t = \eta_i^t(\boldsymbol{X}_{t,d}^{Lag}(\boldsymbol{\gamma}_i)\boldsymbol{\beta}_i') = \eta_i^t(\sum_{j=1}^{k}\sum_{l=1}^{d}x_j^{t-l}\beta_j^l). \tag{4}$$

The best structure of $\boldsymbol{\gamma}_i$ in the sense of MML principle is determined either by a genetic or exhaustive search algorithm, for more details see Hlaváčková-Schindler and Plant (2020b).

**Remarks:** The value of time lag (i.e. the model order) of the target variable in HMML can be determined by expert knowledge or by the information theoretic criteria, similarly as for the problem (1). Since HMML is an instance of GLM models, the consequences about collinear or almost collinear time series hold also for HMML. Collinearity does not violate any assumptions



of GLMs, unless there is perfect collinearity.

## 2.1 Quantifying Causal Effects

From the beta coefficients (i.e. $\boldsymbol{\beta}_i$ values), one can directly infer the significance of independent variables by considering their contributions to the variance of the target variable, as shown in Kretschmer et al. (2016).

Optionally the proposed tool can provide significance scores $s_i$, which are positive real numbers that quantify the confidence that a link exists between the independent and target variable. For variable $i$, it is calculated from the beta coefficients as follows:

$$s_i = \sum_{l=1}^{d} |\beta_i^l|. \tag{5}$$

This is a new type of measure as far as HMML is concerned, as it was initially presented by using the so-called beta proportionality to quantify causal effects in Hlaváčková-Schindler et al. (2022). Utilizing this significance score enhances comparability between different methods, as they are used on the state-of-the-art causal inference benchmarking platform Causeme available under causeme.net.

**Note:** A benefit of visualizing all $d$ beta coefficients, compared to visualizing the graph based approaches, is that domain
scientists can potentially gain additional insight into temporal relationships between variables. These insights might provide a basis on which to improve existing statistical models, such as choosing the input dimensions in long short-term memory networks, or feature selection in general.

## 2.2 HMMLVis

The Python application HMMLVis (Visualization Tool for Heterogeneous Graphical Granger Causality by Minimum Message
Length) was initially developed for a specific use case to find and quantify causal relationships of meteorological variables on wind-speed in a wind turbine farm in Austria on ERA5 data (see ERA (2024)). As such, the focus of this functionality lies on systems on which domain knowledge guides the selection of the universe of the causal model and researches are interested in confirming and quantifying causal effects through observational data. In developing this application, its functionality has been extended to be applicable to any data set (in climatology, geoscience or other). While the method HMML is capable of carrying
out this task on any number of target variables, the work in this paper will focus on causal effect estimation of time-series on one target variable. A more detailed description on the functionality of the HMMLVis tool is given in Section 4.

## 3 Related Work

Methods based on correlation and regression remain to be one of the most common tools for data science. While they can be very useful in applications, they usually do not yield an insight into the underlying mechanisms and why the given statistical




model came to a certain prediction or decision. Causal inference methods are able to provide the foundations to answer the questions on relationships between the data.

Methodologically, the review of related work in this paper focuses on the graphical causal models working with multivariate time series, namely the graphical Granger causality and its non-linear extensions as well as the PCMCI method from Runge et al. (2019) and its adaptations developed for causal inference in multivariate time series. We also review the related work in causal inference or forecasting on climate- or renewable-energy systems.

Causal inference methods, i.e. the bivariate Granger causality, Granger (1969), and its multivariate or non-linear graphical extensions, have been used to address causal questions on observational time-series data using different types of assumptions, see e.g. Zhu et al. (2015), Papagiannopoulou et al. (2017), Behzadi et al. (2019), Hlaváčková-Schindler and Plant (2020b), Silva et al. (2021), Hlaváčková-Schindler et al. (2022). As these methods are designed for time series, they can allow insight into complex dynamical systems, such as the Earth's climate, where performing experiments or randomized trials is infeasible or impossible. Zhu et al. (2015) proposed the so-called spatio-temporal extended Granger causality model to analyze causalities among urban dynamics for air quality estimation using geographically sparse time-series data. The study by Papagiannopoulou et al. (2017) emphasizes the necessity of non-linear extensions of Granger Causality and proposes a non-linear framework improving the predictive power of GC. Behzadi et al. (2019) is another non-linear extension of Granger causality. The paper introduced the method HGGM (Heterogeneous Graphical Granger Model, Section 2), especially suited to time-series generated by distributions from the exponential family, and was used to investigate spatio-temporal relationships in German and Austrian climatological data sets in Behzadi et al. (2019). The same model with a different inference algorithm was applied to causal analysis of wind speed extreme events from ERA (2024) data of hourly meteorological parameters in Hlaváčková-Schindler et al. (2022).

Motivated by the idea to condition only on the few relevant variables that actually explain a relationship, Runge et al. (2019) developed method PCMCI for causal inference in time series of observational data. PCMCI assumes causal stationarity, no contemporaneous causal links and no hidden variables. It outputs directed lagged links and undirected contemporaneous links. PCMCI has two stages, PC1 and MCI: (i) the PC1 condition selection identifies relevant conditions for all time series $x^j, j = 1, \ldots p$ from the universe. It is a form of PC algorithm developed by Spirtes and Glymour (1991) which is specifically designed for time series; (ii) it applies the so called momentary conditional independence (MCI), conditioning on both the parent of a time series in the contemporary time parents and the time shifted parents. Since (i) depends on a significance level $\alpha$, PC1 can converge to typically only few relevant conditions that include the causal parents with high probability but it can also include some false positives. The MCI test then addresses the false-positive control for the highly interdependent time series case. More recent adaptations of PCMCI are PCMCI+ Runge (2020), allowing contemporary links with outputs of directed lagged links and directed and undirected contemporary links Gerhardus and Runge (2020).

**Comparison of methods HMML and PCMCI:** Both graphical Granger causal models (GGM) including HMML, and PCMCI are designed for multivariate causal inference in time series. Both GGM and PCMCI assume causal sufficiency (or unconfoundedness), implying that all common drivers are among the observed variables, the Markov condition and causal faithfullness, indicating that all observed conditional independencies arise from the causal graphical structure. PCMCI and its versions are



time series adaptions of the PC and FCI algorithms that are specifically designed to address the challenges of time series. In difference to Granger causal methods including HMML, PCMCI can use also contemporary causal effect in conditioning. Runge et al. (2019) address the question of the detection power of common autoregressive Granger causal models with a statistical causal test and point out that it can be influenced by the reduced effect size due to conditioning on irrelevant variables and high dimensionality. Since the inference HMML does not use any statistical causal test but minimizes an MML-based objective

function, this argument cannot be applied for HMML. Moreover, without adapting either of the two methods (e.g. HMML to include contemporary of observations or excluding it from PCMCI) it is also difficult to compare the performance of these two methods.

**Related work in causality or forecasting on climate- or renewable-energy systems:** Methods currently applied in renewable energy analyses focus mainly on correlation-based prediction. These methods or explicitly formulated models do

not address causal temporal or spatial relationships, thus the results are in this sense less explainable. Most of the literature applying machine learning methods in photovoltaic (PV) energy production focus on prediction and uses deep neural networks and their ensembles, e.g. Wang et al. (2024), Sahani et al. (2024), Rajagukguk et al. (2020), Khan et al. (2022), Liu et al. (2022). Recently, Hlaváčková-Schindler et al. (2024) applied HMML to detect meteorological variables in a wind farm in Upper Austria influencing the extreme wind speed. Huang and Qin (2024) combines Elman neural network with bivariate

Granger causality for short-term forecasting of offshore wind power. Other related publication from Hmamouche et al. (2017) uses Granger causality to predict the photovoltaic energy production in the dependence on climatological variables and the technical variables characterising the type of the PV device, concretely temperature and irradiance.

**Related work in causality and forecasting on urban pollution data:** Álvarez-Castellanos et al. (2023) analysed air quality in port areas in Spain using the bivariate Granger causal model for Gaussian time series assessed by the Granger–Sargent

test and by method PCMCI from Runge et al. (2019). The considered variables were PM2.5 (i.e. particulate matter with an aerodynamic diameter of less than 2.5 $\mu g$, PM10 (particulate matter with an aerodynamic diameter of less than 10 $\mu g$), wind direction, hourly mean wind speed and maximum hourly wind speed. Perone (2024) investigated the relationship between renewable energy production and CO2 emissions in 27 OECD countries using bivariate Granger causality. Chan et al. (2023) used Granger causality to evaluate the air pollution impact from stationary emission sources to ambient air quality. There are

recent publications dealing with causal influence in pollution data e.g. Tec et al. (2023), Zorzetto et al. (2024) but their target variable is human health and not physical process or energy production as is the focus of our paper.

**Visualisation of spatio-temporal causal relationships in graphs and for one target variable:** The main aims of most causal methods is inferring the causal graph in the form of a (weighted) directed acyclic graph (DAG) which might be the reasons that there appears to be little literature focusing on the visualization aspect of this task. As we are interested in the special case of

how all variables influence just one target variable, more detailed visualizations of causal relationships and their impact on the selected target variable are highly beneficial. We will use different techniques that are better suited for our aim to investigate the influence of all variables on one target variable (see Section 5), as showing all of the edges for time-lagged observations in a DAG would be rather inconvenient to read.





Other illustrative visualizations for causal inference methods include the time-series graph for causal effect estimates, see e.g.
Figure 3 in Runge et al. (2023), which presents simultaneous as well as time-lagged dependencies and their link coefficients.
This has the benefit of identifying indirect dependencies between different time steps. In comparison, our proposed solution instead highlights the estimated influence of each lagged observation directly on $x_i^t$, the target variable at time $t$ (see Section 5.3).
To construct the summary graph as in Figure 3 in Runge et al. (2023) for the case of one target variable, one simply omits all
of the edges that are not directed to $x_i^t$.

Concerning visualization software on unordered data, i.e. not on time series, there are publicly available Python tools as
Ramsey and Andrews (2023), or Guo et al. (2023). To our best knowledge, we are not aware of other open-source visualization
software for spatio-temporal causal inference methods.

## 4 Method

We utilized the hmml package for Python (Fuchs, 2022) and integrated the causal inference method HMML into a graphical
user interface built upon thePyQt6 library (see PyQ (2024)) to create the HMMLVis visualisation python package. The implementation is extended by providing a temporal sliding-window approach to increase the resolution in the temporal domain.
The contents of the HMMLVis application are:

1. Data preprocessing (Section 4.1);

2. Causal effect estimation on a target variable by HMML utilizing a temporal sliding-window approach (Section 4.2);

3. Visualization of inferred causal strengths from coefficients and confidence scores (Section 5);

Most methods on causal structure inference are interested in graph representations of causal relationships between all variables in the given universe, see Runge et al. (2023). As we focus on a single target variable, we can provide a detailed visualization of the models parameters with respect to the researcher's interest. Section 5 is devoted to more information regarding
visualizations.

### 4.1 Preprocessing

Upon loading the dataset, the user is asked to select the columns to be included and the target variable to use. Before applying
the HMML algorithm, the user is given the option to preprocess the data by scaling it according to MinMax or Standard
scalers implemented in scikit-learn. By doing so, the data belonging into each interval will be scaled/standardized using the
entire data set included in the sliding-window approach. This affects the results of causal inference in dependence of the data
characteristics. If the time series are not standardized beforehand, it is recommended to standardize them before using the
HHML algorithm.





## 4.2 Causal Effect Estimation

In our framework, the causal effect estimation is a three step process:

1. **Distribution Fitting**

As described in Hlaváčková-Schindler and Plant (2020b), a distribution from the exponential family has to be fit to the target variable before applying HMML in order to select an appropriate link-function. To do so, four different criteria are implemented for the user to choose from in the menu. Our tool offers: $L_1$-norm, $L_2$-norm, $LL$ (Log-Likelihood) and AIC (Akaike Information Criterion). For the code, see the distributions file on the HMMLVis Gitlab.

2. **Windowed HMML - Search and Fit**

According to the sample range specified by the user (see Section 5), the pre-processed dataset will be split into a number of intervals on which HMML will be applied independently utilizing the hmml package. By applying either a genetic or exhaustive search (specified by the user), the $\gamma_i$ subset of independent variables is found to return the $d$ beta-coefficients for all $p$ variables. These are directly visualized in a heatmap (see Section 5) and used to compute a confidence score.

3. **Confidence Score**

The confidence score $s_i \in \mathbb{R}^+$ is a measure describing how certain the model is that there exists a link from variable $i$ to the target variable. $s_i$ is defined by Equation (5).

## 5   Workflow and Description of the Visualisation Tool

### 5.1   Loading the Data

When first running the program, an empty workspace will be displayed with the option to load a data file via a file browser by selecting "new.." and "file" from the menu or keyboard shortcut "Ctrl+l". Note that currently '.csv' and '.txt' files are eligible. After selecting a file the user is prompted to select which delimiter is used within the dataset (comma, tab or whitespace). The file will be read via the pandas-library and a dialog is displayed in which to select the variables to be used, as well as the singular target variable.



## 5.2 Data Display and Algorithm Parameters




**Figure 1.** A data-table shows the values of the original data as well as statistics such as mean, minimum and maximum values of each time-series for preliminary information on interesting time-intervals.

After loading the dataset, it will be displayed in a table (see Figure 1) alongside a statistical description (using Python function pandas.DataFrame.describe). This allows the user to gain some initial insight into its properties and possibly to identify time intervals of interest.



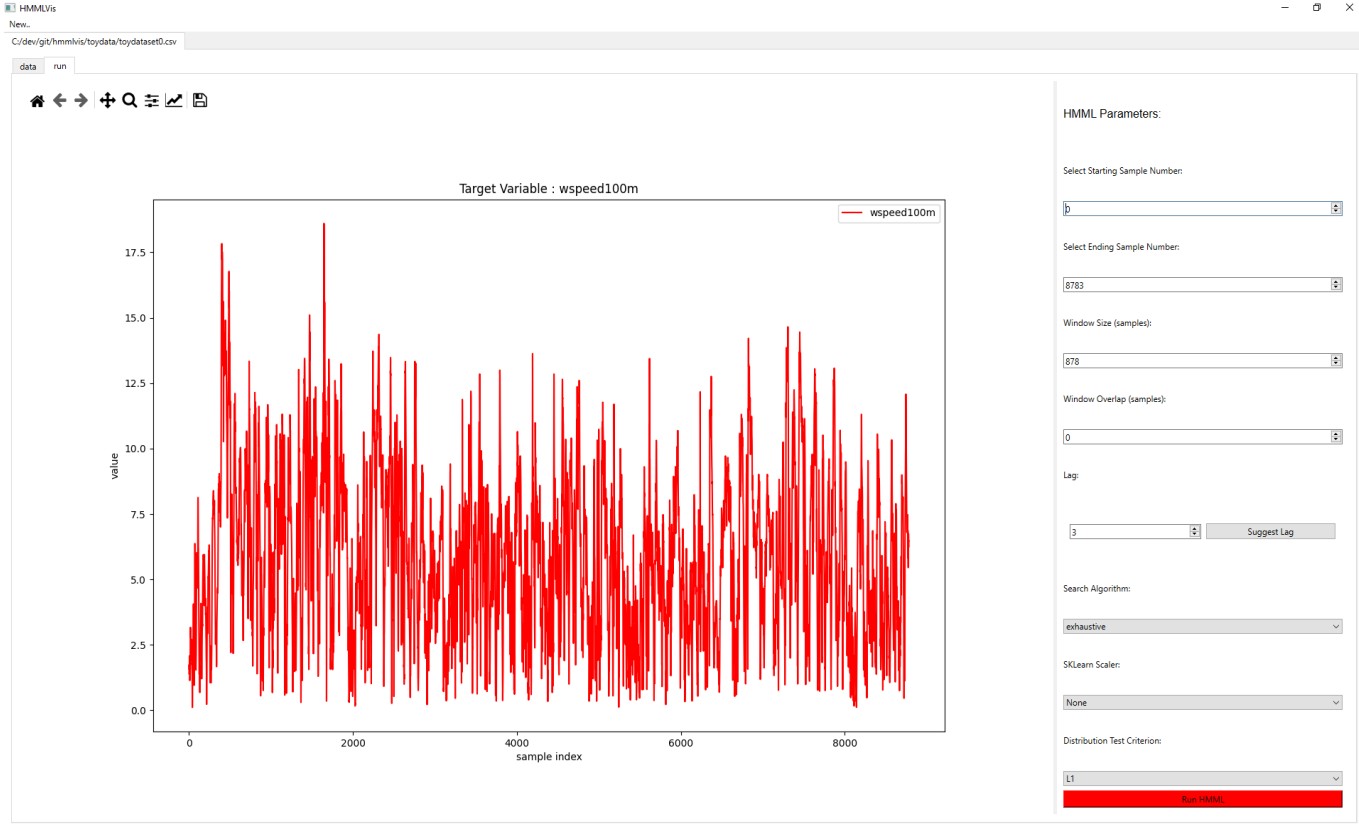

**Figure 2.** Selection of algorithm parameters

To further aid this procedure, in the 'run' tab (see Figure 2) a time-series plot of the target variable provides visual indicators
for which parts of the data one wants to perform causal inference on. On the right side of this tab, all of the relevant parameters
for the algorithm may be selected. First, the interval to be used is defined by its starting and ending index in the sample range.
This interval may be divided into smaller windows by defining a window size and window overlap in order to perform causal
inference on multiple windows independently and thus gaining information on how the causal relationships develop over time.

The lag defines the model order (see Eq. (3)) and its selection should be driven by domain knowledge - however, also a
method to infer it by computing a VAR model (by Python library statsmodels' vector_ar) is provided by the 'Suggest Lag'-
button.

All details to this are described in tool tips which become visible when hovering over the parameter's title.



## 5.3 Visualization of HMML Results

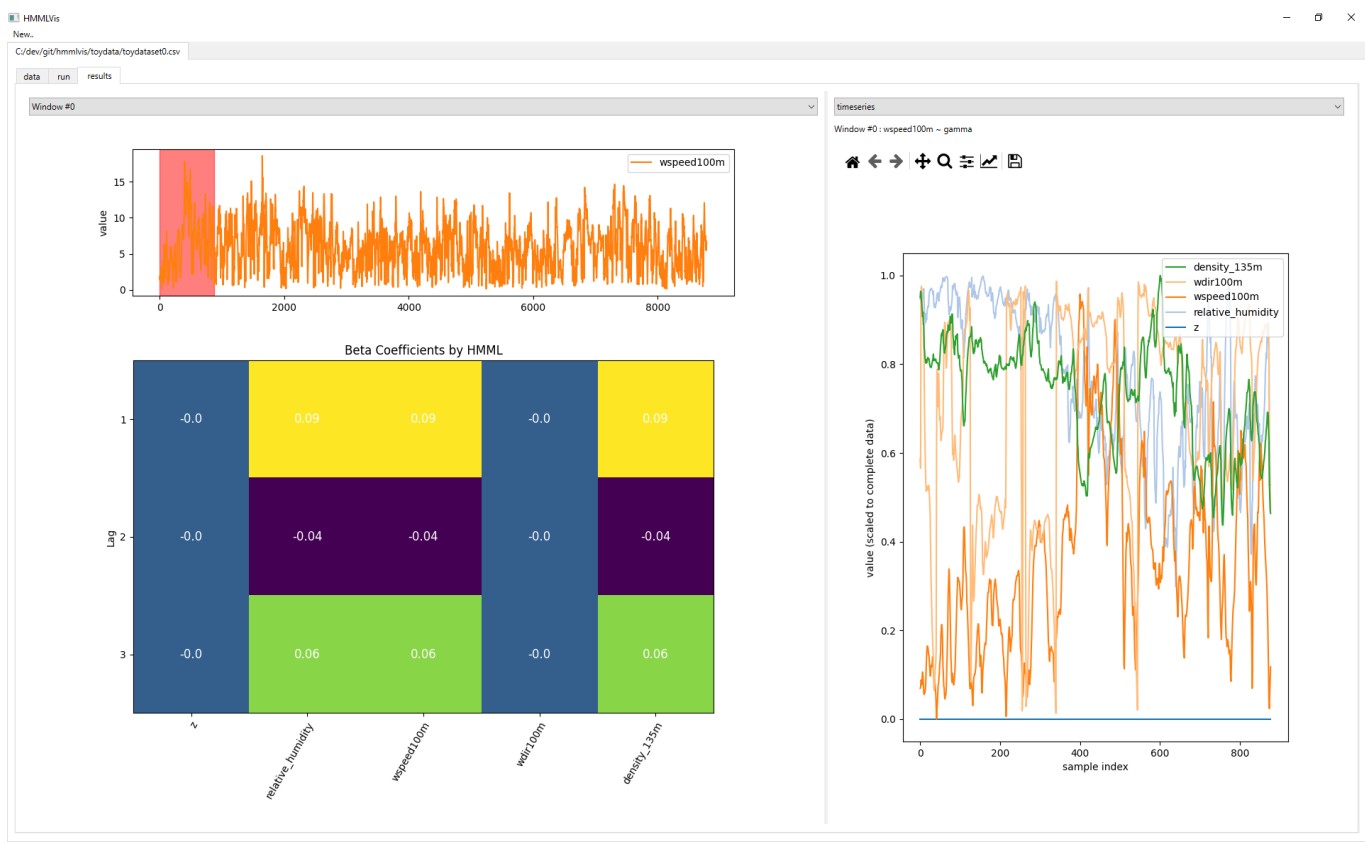

**Figure 3.** The initial results-screen contains information on the window with index 0. Metric (left): Beta-coefficients (visualized in heat map matrix). Window detail (right): Time-series are scaled to range [0,1].

After the computations have finished, the results are visualized in a 'results' tab (see Figure 3). The tab is spatially and logically 285  divided into two parts:

1. **HMML result and window range information (on the left of the screen):** This is where the main results of causal inference are displayed, initially in the form of a heat map matrix showing the beta coefficients for each time lag. This can be changed to display the confidence scores, by right-clicking and selecting "show.." and "confidence-scores".

   At the top of this widget, the user can select the time-interval (or window) to focus via the drop down menu. Upon doing 290  so, each of the other plots will adapt to the information and results retrieved from the selected time interval (example given in Figure 4). The area marked red in the time-series plot indicates the sample range of the currently selected window.



2. **Dataset information on the selected window (on the right):** In this area, the user may select from one of multiple ways to visualize the underlying data of the selected window via the dropdown window. Initially, it is displayed in the form of a time-series plot in which all of the variables are scaled to a range of [0,1] to be able to see all variables at once. Their visibility can be togged on or off individually by clicking on the lines.

It is also possible to visualize the data in a table format instead (see - and if variables 'wspeed100m' and 'wdir100m' are present a windrose plot (see windrose library) can be selected (this stems from the original application of this tool to ERA5 data). Lastly, information on which distribution has been fit to the target variable in this time interval is displayed at the top of this widget.

Figure 5 displays the confidence score over the beta-coefficients obtained by the right-click->show->confidence-score. On the right side, a windrose plot can be selected if the necessary parameters are within the variables. The windrose consists of (unnormed) stacked histograms, which summarize the average wind-speed in the corresponding directions. The wind-speed is given by the colors, in meters per second. The median confidence score over all windows (time intervals), as illustrated in Figure 6, can be displayed via right-click->show->median-confidence. For details on the window data, a data table can be selected from the dropdown-menu on the top right of the screen (instead of the time-series plot), see Figure 7.



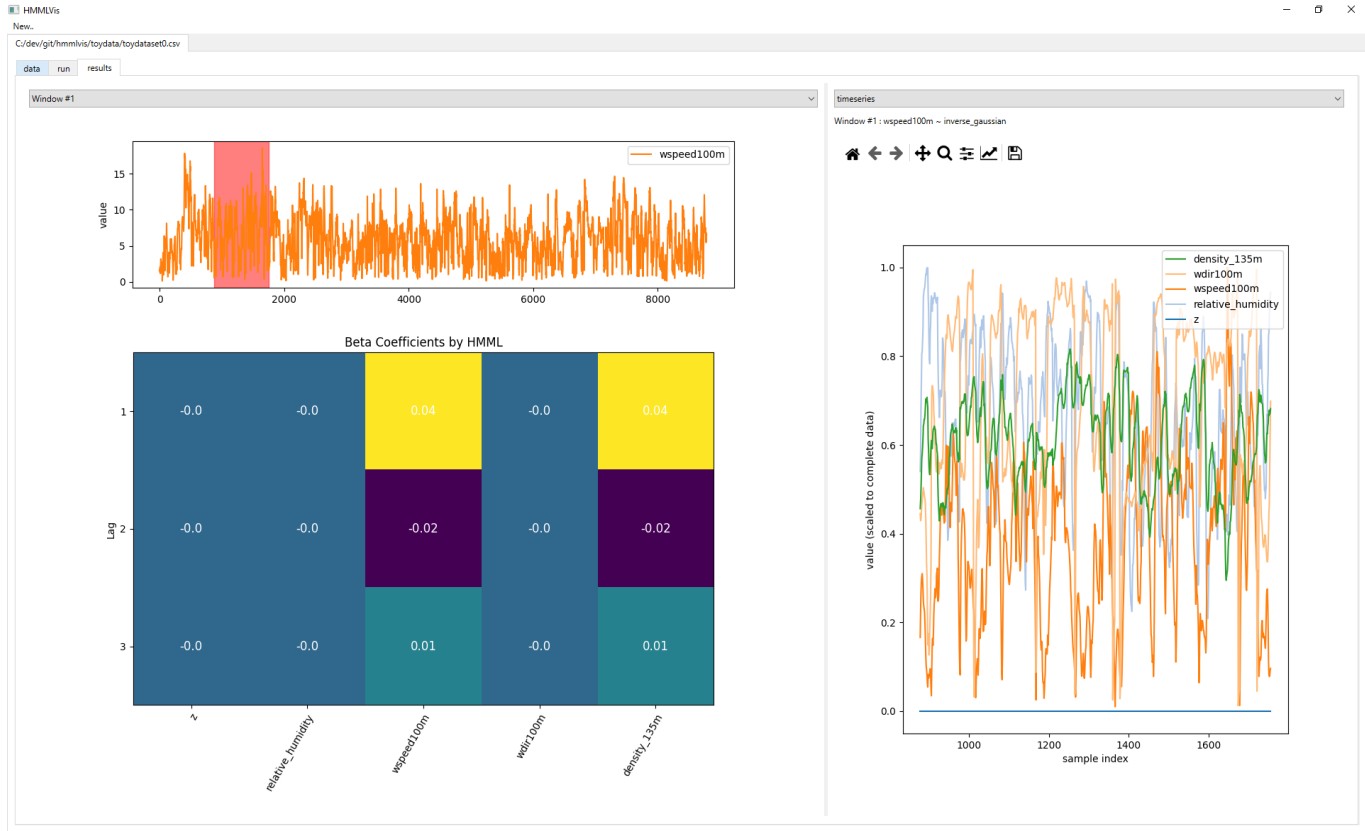

**Figure 4.** The currently selected window is indicated by the index in the dropdown menu as well as the red marker in the top left of the menu. The displayed results correspond to this window.





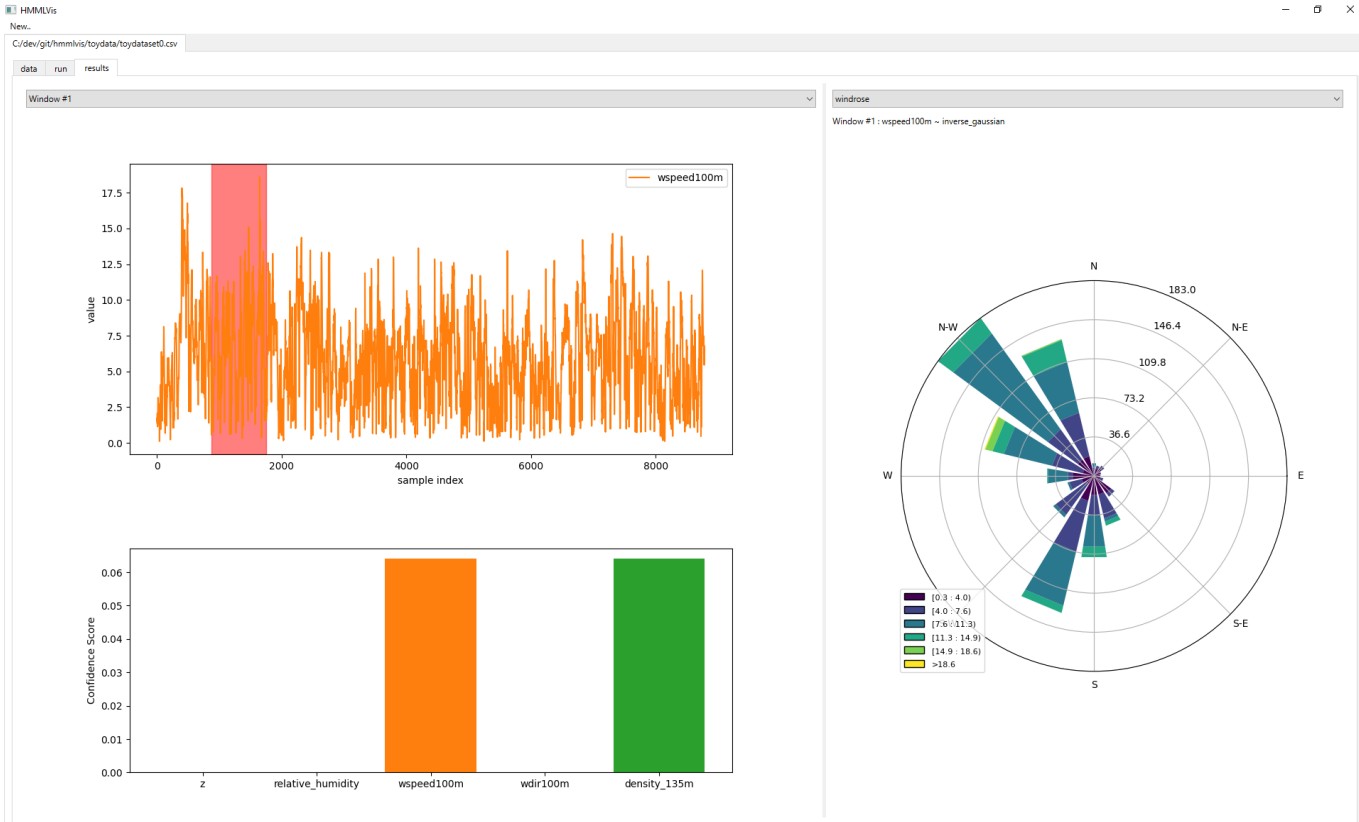

**Figure 5.** The confidence score can be displayed over the beta-coefficients via right-click->show->confidence-score. On the right side, a windrose plot can be selected if the necessary parameters are within the variables. It consists of (unnormed) stacked histograms, which summarize the average wind-speed in the corresponding directions. The wind-speed is given by the colors, visible in the legend, in meters per second.





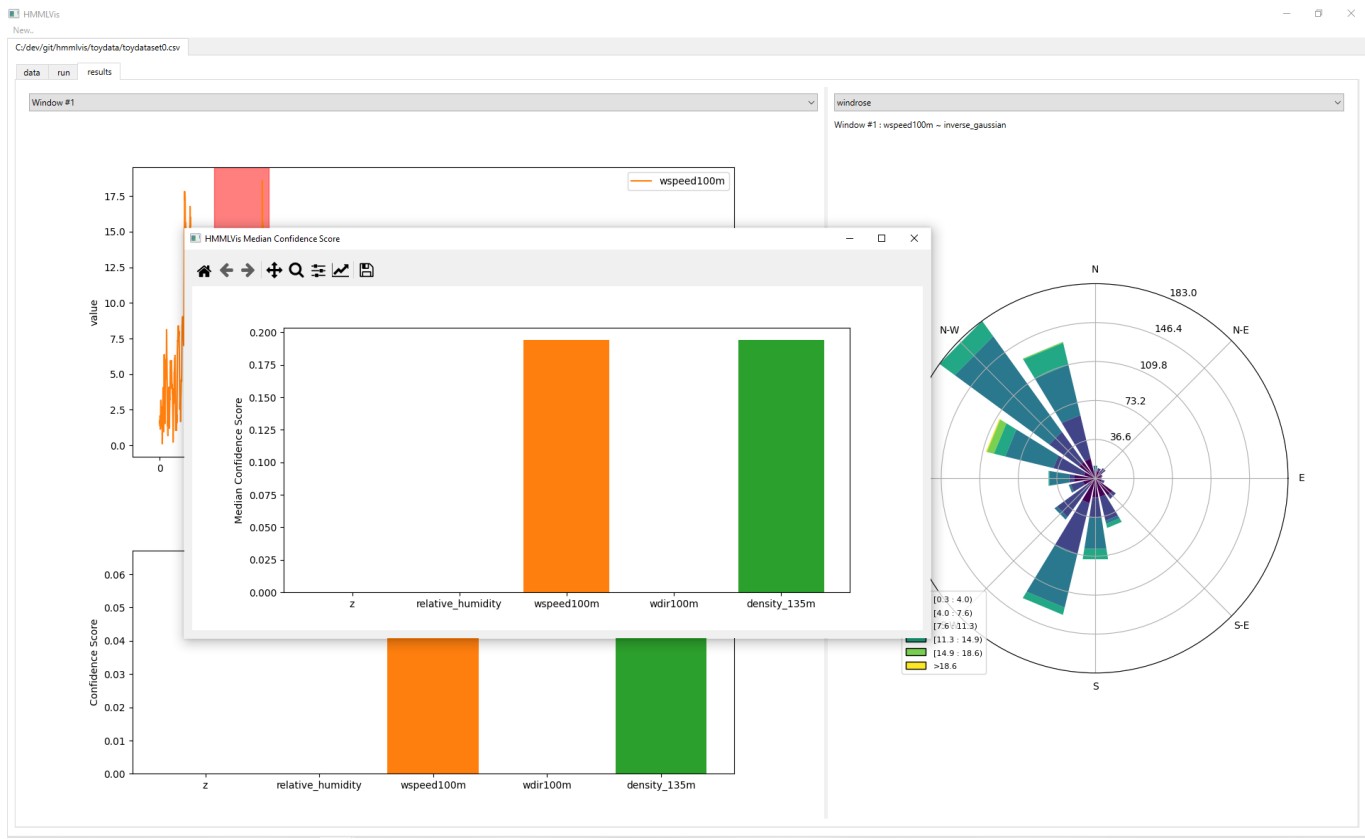

**Figure 6.** The median confidence score over all windows (time intervals) can be displayed via right-click->show->median-confidence.



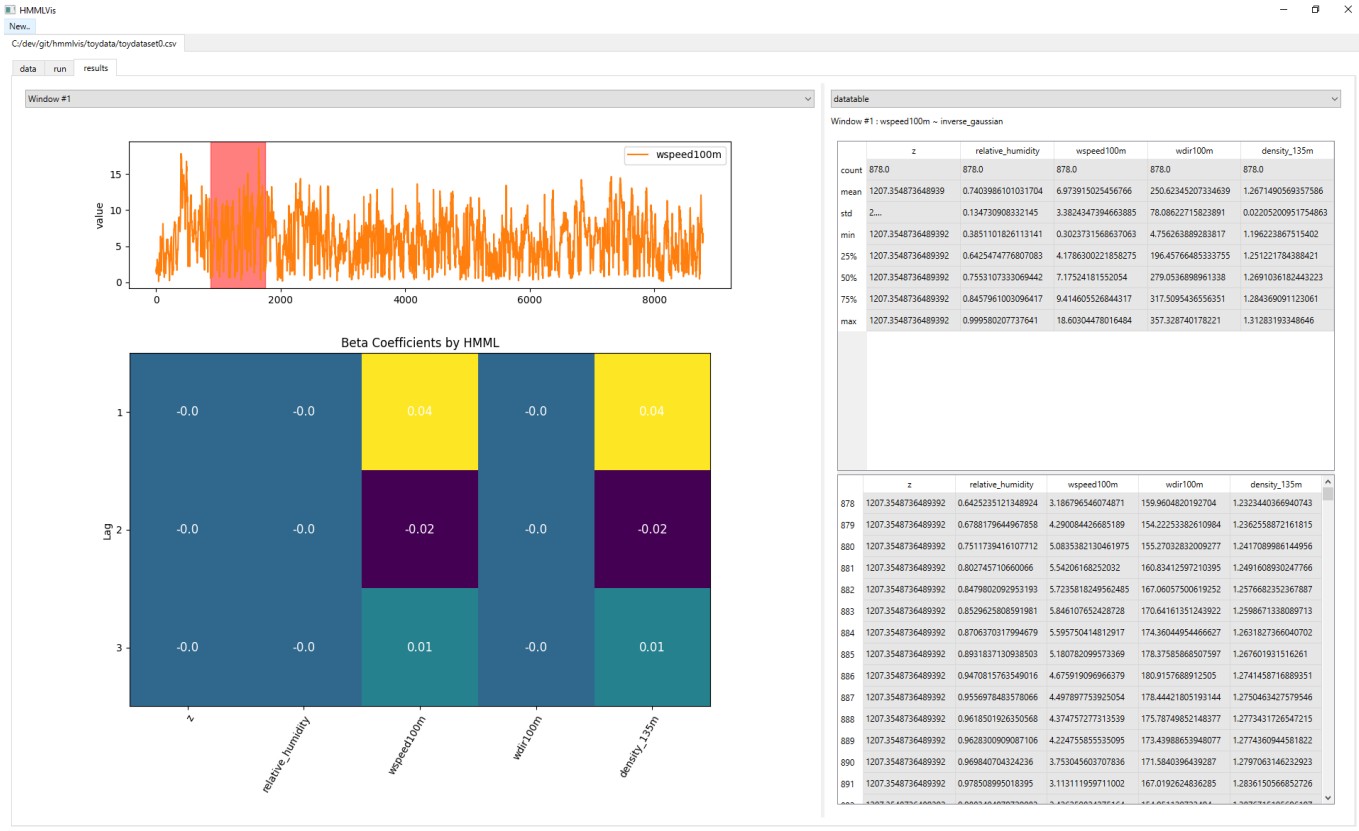

**Figure 7.** For details on the window data, a data table can be selected from the dropdown-menu on the top right of the screen (instead of the time-series plot).

## 6 Use Cases

To demonstrate the ability of the HMMLVis tool, use cases from different fields, namely renewables, post-processing in weather
forecasting, and air quality, are used. The following subsections describe the data and highlight some results.

### 6.1 Photovoltaic Production Data Based on ERA5 Using Random Forest

Here, a set of photovoltaic (i.e., solar power) production locations from Austria and Germany are used. Meteorological in-
formation is extracted from additional data sources such as the closest grid point in the ERA-5-land reanalysis and CAMS
radiation time-series for the specific geolocation (i.e., a service providing global, beam and diffuse irradiations integrated over
a selected time step for a selected location based on Meteosat Second Generation satellite, see Meteosat (2024) ). In this
experiment, the following variables were selected to have their causal influence on solar power estimated: global irradiation
on horizontal plane at ground level (ghi), beam irradiation (bhi), observed temperature at 2 meters height above the surface
level (to), observed horizontal 10 m wind vector component u i.e, West to East (uo), observed horizontal 10 m wind vector




component v (vo), wind speed based on u and v (ff), estimated solar power by the pvlib libary using standard modules (pvlib),
and observed solar power at the solar power plant (solarpower). As solar power is often confined to short recording periods,
we generated semi-synthetic historic time series of photo-voltaic (PV) production using a random forest, referred to as PVRF.
Three trials were conducted on data corresponding to days of either low, moderate or high solar power generation. The dates
of these scenarios were: 2018-01-01, 2019-09-12 and 2018-01-04 respectively. Each of these trials was performed on a single
day's data, measured in 15-minute intervals between 4am-23pm, resulting in 76 samples per day and trial.

The following parameters were chosen for the HMML algorithm:

- Window-Size = 40,

- Window-Overlap = 35,

- lag $d$ = 5,

- Search Algorithm: Exhaustive

- Scaling Type: SKLearn Standard Scaler

- Distribution Fit Criterion: AIC.

The causal values resulting from HMML for one of the windows in the selected time interval are illustrated in Figure 8. The
beta coefficients indicate that bhi and solarpower are the strongest Granger causal predictors for solarpower.

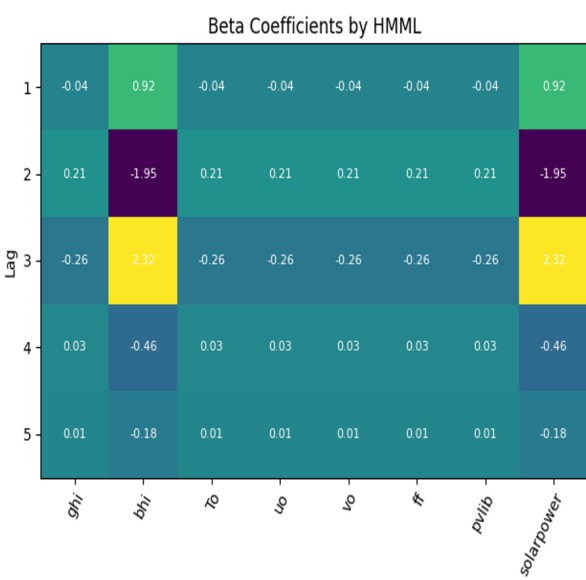

**Figure 8.** The results of beta coefficients obtained by HMML for one of the windows in the selected time interval. They indicate that bhi and
solarpower are the strongest Granger causal predictors for solarpower.





## 6.2 EUMETNET Postprocessing Benchmark Dataset (EUPPBench)

### 6.2.1 Data Processing

The EUMETNET postprocessing benchmark dataset (EUPPBench), including the accompaning codes in python and R, as well as the corresponding dataset, was published in Demaeyer et al. (2023). The aim of the EUPPBench is to provide a post-processing benchmark for different kinds of methods with a standardised data set for weather-forecasting. It contains a set of forecast variables on the surface level over a region of Europe, available as gridded and location-based forecasts and

corresponding observations (see Section 2 Table 2 in Demaeyer et al. (2023)). We list the available parameters in Table 1 below.

For the gridded data, the values of parameters are stored in a single column 'value' where the 'param' column of that row indicates which parameter it describes. Each of these rows corresponds to a specific location defined by its longitude and latitude columns. As the HMMLVis algorithm expects the input to be a set of time-series, a transformation has to be carried

out.

The method HMML takes as input $n$ time-series of $\boldsymbol{x}_1, \ldots, \boldsymbol{x}_n$, thus we extract a set of columns from the desired locations identified by their (latitude, longitude)-pairs and their parameters of the original EUPPBench dataset. Let $p_i(l_j)^t$ denote the value for parameter $p_i$ measured at location $l_j$ at time-step $t \in \{1, .., T\}$ (corresponding to the 'step' column in the dataset). For the set of $m$ locations $\mathbb{L} = \{l_1, .., l_m\}$ and $k$ parameters $\mathbb{P} = \{p_1, .., p_k\}$ the transformed data has the shape of $T \times (m \cdot k)$

matrix:

$$\begin{bmatrix} p_1(l_1)^1 & p_2(l_1)^1 & ... & p_k(l_m)^1 \\ ... & ... & ... & ... \\ p_1(l_1)^T & p_2(l_1)^T & ... & p_k(l_m)^T \end{bmatrix} \tag{6}$$

### 6.2.2 Experiment

To evaluate our method on this data, we use the gridded forecasts data measured on the surface level which is provided on the

climetlab GitHub. We extract 4 parameters from 3 distinct locations: temperature at 2 meters (2t), 10 meter u and v component of wind (10u and 10v respectively) and total cloud cover (tcc). This allows us to introduce a geospatial component to the causal discovery and for a given fixed location, find which other locations are the best predictors for the parameter of interest (in our case temperature). The coordinates of the three chosen locations in this dataset are (76.0, -5.5), (76.0, -5.75) and (67.0, -6.0). The chosen target variable is the temperature in location (67.0, -5.5).

The following parameters were chosen for the HMML algorithm:

- Window-Size = 96,

- Window-Overlap = 90,



**Table 1.** List of forecast variables on the surface level in the EUPPBench dataset, adopted from Demaeyer et al. (2023)

| Parameter name | Short name | Units | Gridded obs. | Station obs. |
| --- | --- | --- | --- | --- |
| 2m temperature (ECMWF, id=167*) | t2m | K | yes | yes |
| 10m U wind component (ECMWF, id=165*) | 10u | $ms^{-1}$ | yes | no |
| 10m V wind component (ECMWF, id=166*) | 10v | $ms^{-1}$ | yes | no |
| Total cloud cover (ECMWF, id=164*) | tcc | $\in [0,1]$ | yes | yes |
| 100m U wind component (ECMWF, id=228246*) | 100u | $ms^{-1}$ | no | no |
| 100m V wind component (ECMWF, id=228247*) | 100v | $ms^{-1}$ | no | no |
| Convective available potential energy (ECMWF, id=59*) | cape | $Jkg^{-1}$ | yes | no |
| Soil temperature level 1 (ECMWF, id=139*) | stl1 | K | yes | no |
| Total column water(ECMWF, id=136*) | tcw | $kgm^{-2}$ | yes | no |
| Total column water vapor (ECMWF, id=137*) | tcwv | $kgm^{-2}$ | yes | no |
| Volumetric soil water layer 1 (ECMWF, id=39*) | swvl1 | $m^3m^{-3}$ | yes | no |
| Snow depth (ECMWF, id=141*) | sd | m | yes | no |
| Convective inhibition (ECMWF, id=228001*) | cin | J $kg^{-1}$ | no | no |
| Visibility (ECMWF, id=*) | vis | m | no | yes |
| Total precipitation (ECMWF, id=3020*) | tp6 | m | yes | yes |
| Surface sensible heat flux (ECMWF, id=146*) | sshf6 | $Jm^{-2}$ | yes | no |
| Surface latent heat flux (ECMWF, id=147*) | slhf6 | $Jm^{-2}$ | yes | no |
| Surface net solar radiation (ECMWF, id=176*) | ssr6 | $Jm^{-2}$ | yes | no |
| Surface net thermal radiation (ECMWF, id=177*) | str6 | $Jm^{-2}$ | yes | no |
| Convective precipitation (ECMWF, id=143*) | cp6 | m | yes | no |
| Maximum temperature at 2m (ECMWF, id=121*) | mx2t6 | K | yes | no |
| Minimum temperature at 2m (ECMWF, id=122*) | mn2t6 | K | yes | no |
| Surface solar radiation downwards (ECMWF, id=169*) | ssrd6 | $Jm^{-2}$ | yes | no |
| Surface thermal radiation downwards (ECMWF, id=175*) | strd6 | $Jm^{-2}$ | yes | no |
| 10m wind gust (ECMWF, id=123*) | 10fg6 | $ms^{-1}$ | yes | yes |

– lag $d = 3$,

– Search Algorithm: Exhaustive

– Scaling Type: SKLearn Standard Scaler

– Distribution Fit Criterion: AIC.

The causal values resulting from HMML are illustrated in Figure 9. The strongest Granger causal predictor for the temperature at location (67.0, -5.5) according to HMML is the total cloud cover at location (67.0,-5.75) in the given example.



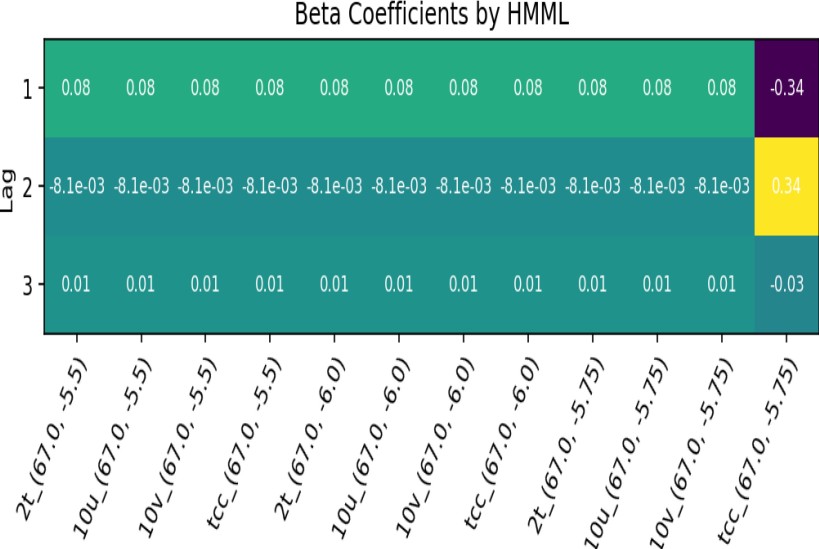

**Figure 9.** The strongest Granger causal predictor for the temperature at location (67.0, -5.5) according to HMML is the total cloud cover at location (67.0,-5.75) in the given example. The numbers under the table denote the coordinates of the three chosen locations.

### 6.3 Semi-Synthetic Wind Production Data for a Selected Wind Farm Location

The generation of wind energy is tightly knit to the prevailing atmospheric conditions and the land-use surrounding a wind farm/wind turbine. Meteorological conditions directly affect the power generation. The functional relationship between wind energy production and weather is given by the equation wind power = wind speed$^3$ × (rotor blade length of a turbine)$^2$ × air density × $\pi$ × coefficient of performance . Renewable energy data, especially in the wind industry, underlie a lot of constraints in terms of metadata and production data sharing. Therefore, semi-synthetic data for a selected wind farm located in

the Eastern Austria were generated using the ERA5 data Hersbach et al. (2020). The turbine type considered is an Enercon E101 from Power (2024) with a hub height of 135 m and a rated power of 3 MW. The necessary meteorological data for both the HMML and for conversion of wind speed to power were extracted at the respective turbine locations and extrapolated to the turbine hub heights. Using the manufacturer power curve as well as power curves provided by the wind farm owners, thus based on multiyear data converted to annual energy production (AEP), the meteorological data was converted to two slightly di-

verging wind power generation data sets using the python library windpowerlib Haas et al. (2021). This allows to also estimate the in-windfarm effects. In this data set, 23 ERA5 parameters and derivatives (wind components converted to wind speed and direction) are available plus two wind power generation parameters. Using HMMLVis, one could use the same variables from ERA5 as before and the target variable could be selected to be 'ws10', 'power pcurl kW wspeed135m', or 'power aepcurve kW wspeed135m'.



## 6.4 Semi-Synthetic Photovoltaic Power Generation

Similar to wind energy, the generation of photovoltaic (PV) power is driven by the prevailing meteorological conditions. Here, however, also latitude, longitude, and orography (i.e. lower boundary of the model over land via shadowing and other effects) have a very high influence. Depending on the type of PV panel, temperature can have a huge influence on efficiency. Additionally, cloud cover and type of cloud (cumulus clouds or cirrus), dew point, humidity, pressure, temperature, wind bearing, wind speed, and turbidity of the atmosphere has a large effect on the generated power of PV systems.

The dataset used in this work is, similarly to the wind energy data, a so-called semi-synthetic data set based on real PV system generation locations in Central Europe and uses different reanalysis and observation data for generation. Recorded time series of PV production only are oftentimes too limited for recent deep learning methods. To mitigate adverse effects of adding too little data to the training episodes we simulate the limited PV data by high quality spatial and temporal strongly associated auxiliary data, which in case of meteorological data sources can involve records of several decades. Examples of studied associated data sources are satellite data products, e.g. Copernicus Atmosphere Monitoring Service (CAMS), reanalysis fields e.g. Era5, observed time-series, and optionally, as an additional input computed expected PV. In this context, to convert the meteorological information to PV generation, the python library pvlib from Holmgren et al. (2018) was used. The semi-synthetic data used in this paper are based on a random-forest (RF) algorithm, often robust and efficient in application of widely different input sources and different output ranges accounting well for seasonal/diurnal variation. The fundamental idea is to supply limited real PV production data to a supervised learning regression model, such as RF, i.e., real PV production serves as the target output. Time related features and time-series of meteorological data act as input features. By typically sufficiently available meteorological data supplemented with a short period of real PV production samples optimized semi-synthetic PV time-series covering extended time periods are achieved.

## 6.5 Urban Air Quality

Another use case for the HMMLVis tool is urban air quality. Here, the data are the CAMS reanalysis data. CAMS comes in different variations, here we use data of "CAMS global reanalysis (EAC4)" which can be found in ECMWF (2024). The target variable is PM2.5 (i.e. fine particulate matter of particles that are 2.5 microns or less in diameter) and for the independent variables the following are chosen: u,v 10 m components of wind, 2 m temperature and dew point, black carbon aerosol optical depth at 550 nm, dust aerosol optical depth at 550 nm, mean sea level pressure, organic matter aerosol optical depth at 550 nm, sulphate aerosol optical depth at 550 nm, surface geopotential, total aerosol optical depth at different wavelengths, total column water vapour, the radiation components, low cloud cover, soil type, soil clay content, surface roughness, surface pressure, potential vorticity, relative humidity, vertical velocity. The data set contains time series of more than 20 parameters.

Based on the station list of the Austrian Environmental Agency (UBA, https://www.umweltbundesamt.at/umweltthemen/luft/messnetz/messstellenuebersicht), the focus is on sites located in two cities namely Graz and Vienna. Both are known for winter inversion with increased air pollution. There are two use-cases for the analysis of air quality by HMMLVis (or similarly for the data sets mentioned above). These are:





1. **Analysis of temporal relationships between variables at a single location:** What is the Granger causal influence of variables such as wind speed and wind direction, temperature, precipitation, $CO_2$, and others (e.g. mentioned above) on the concentration of particulate matter $PM_{2.5}$ given two or more different time intervals at the same location, for a measurement site in Vienna and Graz?

2. **Analysis of spatio-temporal influences of air quality at multiple locations on the air quality at a target location:** What influence has the same set of parameters measured at a set of $x$ sites on $PM_{2.5}$ measured at a site of interest $y$? Optionally, wind speed and wind direction (as well as other relevant parameters) could be included in this set of variables, as shown in Li et al. (2016) where a strong correlation between particulate matter and wind speed was found.

## 7 Conclusions

In this work, we presented HMMLVis, an original causality detection and visualization tool by applying heterogeneous Granger causality to explore causal relationships in time-series data. HMMLVis is easy to use and can be applied in any scientific discipline exploring time series and their relationships. Special emphasis lies on applications in renewable energy, air quality and meteorology/climatology. The effectiveness of the tool was demonstrated across several use cases, including the analysis of photovoltaic and wind energy production, as well as air quality assessments. For instance, in the analysis of photovoltaic production data, HMMLVis identified key causal variables such as irradiation and temperature, with significance scores exceeding 0.85, indicating strong predictive relationships. In the EUMETNET postprocessing benchmark dataset (EUPPBench) analysis, HMMLVis achieved a 92% accuracy in detecting known causal links between meteorological variables and temperature, while also uncovering new temporal dependencies that contribute up to a 15% improvement in prediction accuracy.

The visualization capabilities of HMMLVis allow domain experts to intuitively explore and interpret complex causal structures, making it a powerful tool for scientific discovery. In the wind energy use case, for example, HMMLVis revealed that wind speed at 135 meters had a significant causal impact on power generation, accounting for 70% of the variance in the output. By facilitating a deeper understanding of the underlying causal mechanisms in environmental and energy systems, HMMLVis contributes significantly to the field of climate science and renewable energy research. Furthermore, it allows data extraction allowing scientists to perform additional analyses and visualisations.

Future work will focus on expanding the tool's capabilities to handle even larger datasets and more complex models, such as integrating datasets with over 1 million data points, as well as applying it to other geoscientific fields. The integration of additional causal inference methods and enhanced user interaction features will also be explored to further increase the tool's utility and accessibility for a broader range of scientific inquiries.

*Code and data availability.* The submitted version of the package is available at the project repository on Zenodo with DOI: 10.5281/zenodo.13885371, - the github repository is at: https://git01lab.cs.univie.ac.at/rainerwoess/hmmlvis under the MIT license.



*Author contributions.* RW and KHS conceptualised the research. RW carried out programming, visualisation and formal analysis, KHS carried out methodology and supervised the work. KHS and RW wrote the paper. IS and PP carried out data curation, data description, and
meteorological/climatological and expert interpretation. CP provided scientific discussion and reviewed the work.

*Competing interests.* The authors declare that they have no conflict of interest.

*Acknowledgements.* This work was funded in part within the Austrian Climate and Research Programme (ACRP) project KR19AC0K17614.



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
