# Peer review of "The Spatio-Temporal Visualization Tool HMMLVis in Renewable Energy Applications"

_EGUsphere, 2024_

## Author Comment (AC2)

**Answers to Reviewer 1: The changes have been done into the manuscript.**

1. Scientific Scope and Fit for GMD:

*While causal inference in environmental sciences is a relevant topic, the primary focus of this work is software visualization, and its main contributions lie in user-interface design and graphical rendering of beta coefficients. The paper does not deeply engage with geophysical model development or novel methodological contributions to causal modeling itself. For this reason, the manuscript may be better suited to journals such as EGUsphere preprints, or other open source software journal.*

We thank the reviewer for raising this important point regarding the scope and positioning of the manuscript. We agree that HMMLVis is primarily a methodological and software-oriented contribution; however, we respectfully argue that it aligns with the aims of *Geoscientific Model Development* by supporting the development, evaluation, and interpretation of models applied to geoscientific data.

In the revised manuscript, we have clarified this positioning more explicitly. HMMLVis is not presented as a generic visualization tool, but as a methodologically grounded framework designed to support model-based causal inference in geoscientific applications, including energy meteorology, surface radiation, and urban air quality. The tool builds directly on the heterogeneous graphical Granger model (HGGM) and Minimum Message Length (MML) principles, which are explicitly discussed in the context of geoscientific time series characterised by non-Gaussian distributions, mixed variable types, and limited sample sizes.

We have strengthened this argument by:

1. Expanding the methodological motivation in Section 2.1 (Limitations of Classical Granger Models and Advantages of HMML);
2. Explicitly linking HMMLVis to the evaluation and interpretation of geoscientific models and datasets in Sections 2.2 and 6;
3. Demonstrating its use on multiple geoscientific case studies (wind power, PV power, GHI, and urban air quality), where causal interpretation supports model understanding rather than purely visual inspection.

We believe these revisions clarify that HMMLVis contributes to GMD's core objective of advancing methods for analysing, evaluating, and interpreting geoscientific models and data, while the graphical interface serves as an enabling component rather than the primary scientific contribution.

Manuscript changes:
– Section 2.1 (Limitations of Classical Granger Models and Advantages of HMML)
– Section 2.2 (Model Evaluation)
– Introductory paragraph of Section 6 (Applications)

2. Visualization and Readability:

*Many of the figures (e.g., GUI screenshots, wind rose plots) have awkward scaling or inconsistent proportions, which affects readability. It is recommended to resize and standardize image layouts, especially in Figure 6 and others, to improve visual clarity.*

Figures 1-7 have been recreated to improve visual clarity. In particular, GUI screenshots and plots were resized and standardized to ensure consistent proportions and improved readability.

3. Sections 5 and 6 – Placement:

*These two sections are heavily focused on user-interface walkthroughs and technical instructions. While informative, they resemble a user guide rather than scientific content and would be more appropriate as supplementary material. The main paper should focus on the scientific rationale, methodological innovations, and evaluation results.*

We thank the reviewer for this comment and for highlighting the concern regarding the presentation of Sections 5 and 6. We agree that a clear distinction between methodological description and user guidance is important.

To address this, we have revised the introductory text of Section 5 to explicitly clarify the purpose of these sections. We now frame the workflow description and accompanying examples as a methodological illustration intended to document the data ingestion, causal analysis, and visual exploration pipeline implemented in HMMLVis, rather than as a user guide. The revised text emphasizes that the examples serve to support transparency, reproducibility, and evaluation of the proposed method, in line with the scope of GMD.

Furthermore, Section 6 is now explicitly framed as a set of use cases that illustrate the types of diagnostic insights that can be obtained when applying the tool to different meteorological datasets, rather than as step-by-step instructions.

However, we remain open to moving selected technical details to the supplementary material if the editor considers this more appropriate.

We have added text to cover these comments into Sections 2.1. and 2.2. The scientific rationale, methodological innovations and evaluation are addressed. See below:

2.1 Limitations of Classical Granger Models and Advantages of HMML

Method HMML in general has the following advantages. It supports mixed data: HMML can work with both continuous and discrete variables, which is a major advantage over classical Granger causality methods that typically assume all variables are continuous and Gaussian. HMML handles nonlinear and non-Gaussian data. Further, it allows for modelling interactions between variables of different types without needing to transform or homogenize the data. The MML criterion balances model complexity and goodness of fit, helping to avoid overfitting and underfitting. HMML uses MML to infer the optimal graphical structure (i.e., causal graph),

which includes selecting relevant variables and lags without manual tuning. The graphical output of HMML produces interpretable causal graphs. HMML is suitable for moderate high-dimensional data (for p up to 20). Since some climatological processes are better fitted by exponential distributions than by a Gaussian one, using HMML can be beneficial to inference on our data set. As documented in (Hlaváčková-Schindler and Plant, 2020b), HMML demonstrated significantly higher precision of causal inference regarding the compared methods on synthetic time series, particularly for short time series. This can be relevant in many climatological applications that work with short time series, i.e., time series with a number of observations on the order of up to a thousand times the number of involved processes among which causal inference is sought.

2.2 Model Evaluation

(Hlaváčková-Schindler and Plant, 2020b) examined the precision of the output causal graphs obtained by HMML in comparison to benchmark methods on synthetic data, where the ground truth, i.e. the target causal graph was known. F1- measure (or F-score, in other words) was used as an evaluation metrics. Randomly generated processes having various exponential distributions were examined together with the correspondingly generated target causal graphs. The performance of HMML, as well as of the benchmark methods HGGM (Behzadi et al., 2019), SFGC (Kim et al., 2011) and Linear Non-Gaussian Acyclic Model, i.e. LINGAM (Shimizu et al., 2006), depends on various parameters including the number of time series (features), the number of causal relations in Granger causal graph (dependencies), the length of time series and finally on the lag parameter. HMML significantly outperformed in F1-measure the comparison methods in all investigated cases. More details to the experiments can be found in (Hlaváčková-Schindler and Plant, 2020b).

*4.   Terminology and Acronyms:*

*The manuscript contains several instances where acronyms (e.g., EUMETNET) are introduced without first spelling out the full name. This is not compliant with standard academic writing practices. Please ensure that all acronyms are introduced in full upon first use, followed by the abbreviation in parentheses.*

We have explained the acronyms in full words upon their first use.

5.   *Motivation for HMML and MML:*

*The rationale for using heterogeneous Granger models and MML-based feature selection should be articulated more clearly for readers unfamiliar with these frameworks. What concrete limitations of classical Granger models does HMML overcome, especially in environmental data contexts?*

We have added text to cover these comments into Section 2.1 (see the text above in Comment 3).

*6.   Model Evaluation:*
*While the tool is applied across several datasets, the evaluation lacks clear performance metrics or validation benchmarks. For example: How does the tool's output compare with known or simulated*

*ground-truth causal structures? Are the inferred relationships stable across time windows and locations? Could precision/recall, consistency, or information gain be reported?*

We have added text to cover these comments into Section 2.2 (see the text above in Comment 3).

*7. Synthetic Data Use:*

*The paper uses semi-synthetic datasets (e.g., for PV and wind), but the construction process needs to be described in more detail. How realistic are these datasets? What modelling assumptions underlie their creation, and what uncertainties are introduced?*

We thank the reviewer for this important comment. We agree that the original manuscript did not sufficiently explain how the semi-synthetic datasets were constructed, validated, and limited in scope. In the revised manuscript, we have substantially expanded the description of all semi-synthetic datasets, clarified their purpose, and added an explicit validation figure comparing semi-synthetic and operational data. Specifically, we have introduced a dedicated subsection in the Data and Methods section (Section X.X: Semi-synthetic datasets) and added a new multi-panel figure (Fig. X) that directly compares semi-synthetic and measured time series for wind power, photovoltaic (PV) power, and global horizontal irradiance (GHI).

Wind power (onshore wind farm)

For the wind-energy case study, we start from ERA5 reanalysis wind fields and downscale them to turbine hub height at the wind-farm locations using standard vertical interpolation and site-specific adjustments. The resulting hub-height wind speeds are converted to electrical power using manufacturer-provided power curves and the installed nominal capacity of each turbine.

The resulting ERA5-synthetic daily power time series is compared against anonymized operational turbine data from the same wind farm for the period 2016–2020. Daily values are used, days with obvious curtailment plateaus are excluded, and both measured and semi-synthetic series are normalized to the range [0–1] for anonymization. Figure X (left column) shows normalized monthly means, daily scatter plots, and probability density functions. The correlation between daily measured and ERA5-synthetic wind power is r ≈ 0.91, and the distributions agree well over most of the range, indicating that the semi-synthetic data reproduce the observed daily-to-seasonal variability sufficiently well for the purposes of this methodological study.

Photovoltaic (PV) power

For the PV case study, semi-synthetic PV production is constructed using ERA5 meteorological fields combined with CAMS radiation and atmospheric composition products. These inputs are processed through a simple PV performance model using known plant characteristics (installed DC capacity and orientation). Daily semi-synthetic PV power is compared against measured plant output, again normalized to [0–1]. As shown in Fig. X (right column), the daily correlation between measured and ERA5+CAMS-synthetic PV power is r ≈ 0.98, and both seasonal cycles and daily variability are closely reproduced.

Global horizontal irradiance (GHI)

For GHI, ERA5+CAMS semi-synthetic irradiance is compared to long-term measurements at a radiation reference station. The daily correlation is r ≈ 0.99, and the probability distributions are nearly indistinguishable (Fig. X, middle column), demonstrating that the semi-synthetic irradiance series provide a highly realistic representation of observed surface radiation conditions at this site.

Scope and limitations

We now explicitly state in both the Introduction and Discussion that the semi-synthetic datasets are designed to be realistic but not perfect representations of operational data. Their purpose is to provide controlled, anonymized, and physically plausible time series for demonstrating HMMLVis, rather than to quantify the exact performance of individual energy systems. Uncertainties arise from reanalysis biases, simplified power-conversion models, and residual curtailment or data-quality effects, and these limitations are discussed in the revised manuscript.

Overall, the expanded description and the new validation figure clarify how the semi synthetic datasets are constructed, document their realism with respect to available operational data, and clearly delimit the scope of the conclusions drawn from these case studies.

Changes in the manuscript:
1. New subsection additions to Section 6.1 and 6.3. describing the synthetic data generation
2. New Figure 8 comparing measured and semi-synthetic wind power, PV power, and GHI
3. Additional clarifying sentences in the Introduction and Discussion on the scope and limitations of semi-synthetic data

*8. Link Functions and GLMs:*

*More explanation is needed regarding the choice of link functions and distributions within the exponential family. Were they selected empirically, or based on expert input? Was model fit compared across different options?*

To explain this, we added the following text into Section 2 before Eq.(2): The best- fitting distribution of the target time series within each interval can be identified using the Residual Sum of Squares (RSS) and the Kolmogorov–Smirnov (K–S) test.

*9. Scalability and Runtime:*

*Please provide information on the computational cost of using HMMLVis, particularly with sliding windows and larger variable sets. How long does one window take to process?*

We added subsection "Scalability and Computational Complexity of HMML" where we present the theoretical upper bound on the complexity of method HMML into the manuscript. We introduced a subsection on runtime 6.5., which includes Table 6 summarizing the per-window runtime for each dataset for which results are presented.

**Minor Comments**

*1.   Throughout the manuscript, some notations (e.g., $\beta\backslash beta\beta$, $\eta\backslash eta\eta$, indices $i,j,t i,\ j,\ t i,j,t$) are inconsistently formatted—sometimes in plain text, sometimes in math mode. Ensuring typographic and notational consistency across all equations and text would improve professionalism.*

We have corrected this in the manuscript.

*2.   While most figures are labelled, some captions could be more descriptive to help readers interpret them without referring to the main text. For example, indicate clearly what variables or locations the time series refer to, what colour scales represent, or whether the visualizations correspond to real or synthetic datasets.*

We updated description of all figures in Sections 5 and 6 to make them more readable without referring to the main texts.

*3.   The abstract currently blends methodology and application without clearly delineating the main contribution. Consider restructuring it into: (1) motivation, (2) method, (3) key results, (4) broader implications—to improve clarity and impact for readers scanning the abstract alone.*

We restructured the abstract based on your suggestion.

*4.   Some sentences are overly long or have ambiguous phrasing, particularly in Sections 2 and 4. For instance, compound sentences mixing mathematical definitions and explanatory text can be split for better readability. A careful language edit would improve clarity.*

We have carefully revised Sections 2, 4, and 5 to improve their readability and clarity.

*5.   Check reference formatting for consistency (e.g., Behzadi et al. (2019) vs. Behzadi et al., 2019). Ensure all references are cited in a consistent style and match the GMD citation standards.*

We made the reference formatting consistent in the whole paper.

---

## Author Comment (AC3)

**Answers to Reviewer 2: The changes have been done into the manuscript.**

1. *The manuscript would benefit from enhanced clarity, organisation, and presentation to align with the publication standards, specifically using a structure Introduction, Methods, Results, Discussion, Conclusion could help in the readability. Also The manuscript introduces several acronyms without providing their full names at first mention.*

We structured Introduction, Methods, Results, Discussion, Conclusion and improved their readability.

*Could you check if The Digital Object Identifier (DOI) 10.5281/zenodo.13885371 of the Code and data availability section corresponds to private Zenodo record that is not publicly available.*

The files are publicly accessible according to zenodo at :
https://doi.org/10.5281/zenodo.13885371

*List of **keywords** is missing after the abstract.*

We added key words into page 1.

*Figure 1 is not relevant information and too small numbers for comfortably reading and also a white lost space between the grey tables is there that could be reduced.*

Figure 1 has been updated to improve readability.

*Figures 2- 7 need to be presented in a more organised way for a publications, those are snapshots of the images.*

All figures in section 5, namely Figures 1 - 7, were updated to improve readability.

*Figure 2 the caption of the image needs to be complemented, more description to be added.*

More detailed descriptions were added to figures, with a focus on Figures 2 and 3 introducing the main functionality of the application.

*Figure 3 and Figure 4, use subplots (a) (b) and (c) to introduce each subplot in the caption.*

We introduced separate descriptions for top-left, bottom-left and right subplots in these figures.

*Figure 6 looks a bit as a messy plot, try to make a mosaic with the subimages not overlapping ones to others to illustrate what you want to achieve with this visualisation.*

We separated the image of the median confidence score into Figure 6 (as the figures which were in the background were previously showcased). This plot opens as a separate window in the application.

*Line 325: Make a table or put this list information in a descriptive paragraph*

The parameters used in the given sections were listed in Tables 1 and 3.

*Line 340:*

We list the used parameters used in the experiments section for the datasets in Tables 1, 3 and 5 respectively for photovoltaic production (Section 6.1) , meteorological benchmarking (Section 6.2) and air quality (Section 6.4). We also list the parameters, their abbreviations and units for the air quality dataset in the new Table 4.

*5: 6.2.1 Data Processing section appear in a part which should be the conclusive part of the paper after the results but this start describing something more appropriate from methodology.*

Table 1 --> create a new column with the identificators (ECMWF, id=167*) parts of the parameter name column. Also, in the units some J appears bold and others italic.

The identificators of the variables were separated into their own columns and the units standardized (in what is now Table 2, section 6.1).

*The section 6.5 Urban Air quality miss some result plots or extra information, or this use cases with no discussion just presentation could be briefly presented in the introduction of the manuscript*

We thank the reviewer for this helpful comment. We agree that the Urban Air Quality (UAQ) use case was previously presented too briefly and without sufficient discussion. In the revised manuscript – in Section 6.4, we have clarified the role and scope of the UAQ case study and expanded the accompanying explanation. Specifically, we now:

Explicitly frame the UAQ example as a demonstrator use case for HMMLVis, intended to illustrate the tool's ability to explore heterogeneous causal relations between meteorological drivers and air-quality variables, rather than to provide a comprehensive atmospheric chemistry analysis.

We added the following text:

Urban air quality is influenced by a combination of local meteorological conditions and regional pollutant transport. To illustrate the capabilities of HMMLVis beyond renewable-energy applications, we present urban air quality as a demonstration use case. Specifically,

the tool is applied to explore heterogeneous causal relationships between meteorological drivers and PM 2.5 concentrations in Vienna and Graz. This example is intended to showcase the exploratory and visual analysis functionality of HMMLVis, rather than to provide a detailed or exhaustive assessment of atmospheric chemistry processes.

We added a short interpretative discussion linked to Fig. 12, highlighting the dominant causal links identified by HMMLVis (e.g. temperature, wind speed, boundary-layer stability proxies) and their consistency with established physical understanding of urban air-pollution processes.

We refer to Figure 12 already in the introduction, indicating that UAQ is one of several application examples used to demonstrate the versatility of HMMLVis across energy-meteorology and environmental domains.

We believe these changes improve the clarity of the UAQ use case and align its presentation with the methodological focus of the manuscript.

**Manuscript changes:**
– Introduction: last paragraph of Section **1**, where the UAQ case study is introduced as a demonstrator alongside energy-related applications
– Section **6.5 Urban Air Quality**, where the scope of the use case and the interpretation of Fig. 12 are expanded
– Discussion: short paragraph referencing UAQ as an example of exploratory causal analysis rather than a full atmospheric chemistry study

*References. Needs to add the doi for the papers that it is missing.*

We have done it.